# Characterization of the physical properties of electron-beam-irradiated white rice and starch during short-term storage

Zhihong Du[1,2], Jiali Xing[3,4], Xiaohu Luo[1,2]*, Li Wang[1], Lihong Pan[1], Yulin Li[2], Ren Wang[1], Yuntao Liu[5], Xiaohong Li[6], Zhengxing Chen[1]*

1 National Engineering Laboratory for Cereal Fermentation Technology, Jiangnan University, Wuxi, China, 2 Hubei Key Laboratory of Edible Wild Plants Conservation and Utilization, Hubei Normal University, Huangshi, China, 3 Ningbo Institute for Food Control, Ningbo, China, 4 School of Marine Science, Ningbo University, Ningbo, China, 5 College of Food Science, Sichuan Agricultural University, Yaan, China, 6 Department of Food and Biological Engineering, Beijing Vocational College of Agriculture, Beijing, China

* xh06326@gmail.com (XL); zxchen_2008@126.com (ZC)

**Data Availability Statement:** All relevant data are within the paper and its Supporting Information files.

## Abstract

Electron-beam irradiation (EBI) is a cold sterilization technology used in the irradiation processing of food, including rice. Herein, the effects of EBI on the swelling power, color, pasting, and sensory properties of white rice after short-term storage were analyzed. Samples were electron-beam irradiated at 0, 2, 4, 6, or 8 kGy and stored at 25 ˚C or 37 ˚C for up to 75 days. Results showed that swelling power and major pasting viscosities (including peak, breakdown, and setback viscosities) at both storage temperatures decreased with increased irradiation dose. Negative correlations were also observed between the major viscosities of pasting properties and irradiation dose at both storage temperatures. During sensory evaluation, extremely low scores for rice hardness, appearance, taste, and overall acceptability were obtained for rice subjected to high EBI dose (>4 kGy). However, rice stored at 37 ˚C showed lower performance than rice at 25 ˚C in terms of the abovementioned parameters. By contrast, the sensory properties at irradiation doses between 2 and 4 kGy were better than those of the control group at both storage temperatures. All these findings indicated the potential of low-dose (<4 kGy) EBI as pretreatment for improving the quality of white rice during storage.

## Introduction

Rice is an important staple food in Asia. According to 2017 crop statistics from the Food and Agriculture Organization of the United Nations, rice accounts for approximately 7.4% of all crop production in Asia, with 90% of the world's rice produced in the continent [1]. During prolonged storage, decreased rice quality and eventual rancidity (undesirable color, flavor, and taste) occur, resulting in extensive economic loss [2, 3]. The source of this deterioration is multifaceted, although the development of molds and insect activity are significant contributors [4]. In particular, such negative effects on crops are increasingly intensified under unfavorable

**Funding:** This research was financially supported by National Key Research and Development Program of China (2017YFC1600904, 2017YFD0401200), Jiangsu Agriculture Science and Technology Innovation Fund CX(17)1003, Open Foundation of Beijing Advanced Innovation Center for Food Nutrition and Human Health (20182014), China Agriculture Research System (CARS-02-32), Zhejiang Food and Drug Administration Science and Technology Project (2018) (201802, 201811), Science and technology research and development promotion project of Beijing Vocational College of Agriculture (XY-YT-16-44), National first-class discipline program of Food Science and Technology (JUFSTR20180203).

**Competing interests:** The authors have declared that no competing interests exist.

storage conditions with poor drying treatments, high level of air relative humidity, strong light sources, or high temperatures [5, 6]. Thus, once produced, white rice normally needs to be consumed within several months before significant deterioration occurs. The shelf life of white rice in ordinary packaging is usually restricted to between 6–12 months. The shelf life of white rice at conventional bulk storage is not clearly defined but is usually considered to be 3 months in summer and 6 months in winter or autumn under optimal storage conditions. However, when white rice (especially those in bulk), is stored in open and uncontrolled environments, a considerably shorter shelf life than normal is expected.

Multiple treatments targeting molds and insects have been studied to extend the storage period of rice, with various types of irradiation being a common subject. These processes can exert significant reproduction–inhibition and inactivation effects on most microorganisms and insects by damaging cellular structures or disturbing the physiological functions of their living cells [7–9]. For example, an electron-beam irradiation (EBI) dose of 1 kGy can kill some pestilent insects and their eggs, such as rice weevil, red flour beetle, and *Rhizopertha dominica*, in grains during storage. Increased irradiation dose of 4 kGy can inactivate up to 99% of microorganisms [10]. Remarkably high doses are not recommended for phytosanitary purposes because these doses can considerably affect cereal quality and are usually adopted for investigating physicochemical changes of these samples. Irradiated foods are also not widely accepted by most consumers because of their safety uncertainty. Foods irradiated 10 kGy do not pose any toxicological, microbiological, or nutritional problems according to the World Health Organization [11].

EBI is a cold sterilization technology with evident advantages in food irradiation processing relative over other methods. The high-energy electron beam generated by an electron accelerator has a large beam power and rapid processing speed. The beam can also be efficiently utilized because the energy released by EBI is highly concentrated and always in the same direction, thereby effectively preventing energy loss. Apart from the convenient operation of an accelerator, its beam intensity can be tailored to various sizes, types, and quantities of cargo. Compared with gamma irradiation, whose safety problems are prominent, an EBI accelerator is relatively safe to use because its ray emission can be completely arrested whenever needed [12].

Unfortunately, studies on the effects of EBI on white rice during short-term storage and its proper irradiation dose are limited. By contrast, the effects of gamma irradiation on the physicochemical properties of grains and starch have been extensively investigated. For instance, the pasting properties, apparent amylose content, gelatinization temperature, swelling power, and texture parameters of grain and starch subjected to increasing gamma radiation doses have been studied and found to decrease [13, 14]. Long-term studies on the physiochemical and sensory properties of gamma-irradiated brown rice stored for 1 and 1.5-year periods have also demonstrated the feasibility of delaying rice deterioration [15–17]. EBI is speculated as an effective pretreatment for maintaining the quality of white rice quality during storage.

Starch is the main component of rice. Its pasting properties may largely represent the rice properties. However, rice and starch may also differ in pasting properties because of their varied nutrient amounts. For example, protein denaturation by heating results in structural changes, which induce hydrophilic groups (e.g.,–OH, −NH$_2$, −COOH, and −SH) to form cross links with the starch matrix and obstruct swelling, leading to viscosity reduction [18]. Therefore, assuming that the pasting properties between rice flour and starch varied, this work aimed to test the hypothesis and determine if the difference can be weakened or enhanced by EBI treatment.

Furthermore, we have previously investigated the sterilization effects of EBI on brown rice and milled rice and proved this method's effectiveness. Results showed that the total viable

bacterial count in brown and milled rice was significantly reduced from approximately $3 \times 10^3$ CFU/g at 0 kGy to 10 and 0 CFU/g at 5 kGy, respectively [19]. Accordingly, this study did not focus on microorganisms and investigated instead the effects of EBI on the pasting, swelling, and sensory properties of white rice within 75 days of storage. The storage temperatures of 25 ˚C and 37 ˚C were selected to emulate the expected temperature range in a storage facility operating under ideal and high-temperature conditions, respectively.

## Materials and methods

### Materials

White rice (Daohuaxiang 2, *Oryza L.*) was bought in a local supermarket and it was cultivated in Harbin City, Heilongjiang Province.

### Irradiation and storage of white rice

Before irradiation, 250 g of white rice was placed in polyethylene bags and sealed for each replicate. The bags were then irradiated at room temperature for 5 s by using an electron accelerator (AB5.0 type) at Wuxi ELPONT Irradiation Technology Co., with irradiation doses of 0, 2, 4, 6, and 8 kGy. The dose was selected on the basis of previous studies [19, 20]. After irradiation, the rice samples were individually stored at 25 ˚C or 37 ˚C for 75 days. During testing, each rice sample was milled and sieved through a 40- and 100-mesh sieve (sieve diameter of 350 and 150 μm, respectively) to determine its water content and color difference and pasting properties and swelling power, respectively. Each sample was irradiated in triplicate.

### Determination of chemical composition of white rice

Protein, total starch, total dietary fiber, fat, and ash of white rice and its water content during storage was determined in accordance with the Association of Official Analytical Chemists standard methods.

### Starch isolation

Starch was isolated following the method described by Zhong et al. [21] with some modifications. White rice (50 g) was softened in distilled water for 4–8 h for wet milling. The slurry was then centrifuged for 10 min at 4000 r/min (LXJ-IIB, Shanghai Anting Scientific Instrument Factory), and the pellet was collected and dispersed in 200 mL of 0.2% aqueous NaOH at ambient temperature for 48 h (aqueous NaOH was replaced once at 24 h). The slurry was then filtered through 100- and 150-mesh sieve (sieve diameter of 150 and 106 μm, respectively), and then centrifuged. After draining off the supernatant, the pellet was washed 3–4 times with distilled water, and the top yellow layer of the pellet was simultaneously scraped off to remove protein. The remaining solids were dispersed in distilled water, and pH was adjusted to 7.0. This suspension was then centrifuged and washed three times with distilled water to wash off NaCl. The clean and white t was collected and dried for 24 h in a convection oven at 40 ˚C, milled, and screened with a 100-mesh sieve to test its pasting properties.

### Measurement of pasting properties

The pasting viscosity of rice flour and starch was determined using a rapid viscosity analyzer (RVA) (RVA 4500, Perten Instruments, Australia) through a method described by Du et al. [22] with slight modifications. Rice flour and starch (3 g) were dispersed in distilled water

(25 g) in an aluminum container. The mixture was then stirred at 960 r/min for 10 s and at 160 r/min for the remainder of the test. Heating was sustained for 13 min beginning at 50 ˚C for 1 min and increased to 95 ˚C within 222 s for 150 s, and then cooled to 50 ˚C.

## Swelling power measurement

The swelling power of rice flour and starch was measured following a method from Tananu-wong et al. with some modifications [23]. The water-bath temperature was selected based on the pasting temperature of rice flour and starch in the RVA results. A flour slurry containing 0.5 g of flour and 15 mL of distilled water was mixed using a vortex oscillator. The flour slurry was heated in a 70 ˚C or 90 ˚C water bath for 30 min. This mixture was cooled to room temperature and centrifuged for 10 min at 4000 r/min (LXJ-IIB, Shanghai Anting Scientific Instrument Factory). The supernatant was dried for 3.5 h at 105 ˚C. The swelling power of white rice flour was calculated in accordance with the following equation:

$$\text{Swelling power(g/g sample db)} = \frac{\text{weight of precipitated paste(g)} \times 100}{\text{sample weight(g db)} \times (100 - \% \text{ solubility})} \quad (1)$$

$$\%\text{Solubility} = \frac{\text{weight of dried supernatant(g)} \times 100}{\text{sample weight(g db)}} \quad (2)$$

## Colorimetric assay

The colorimetric assay of the white rice flour (four spots measurement for each flour sample) was determined in terms of $L^*$ (lightness) and $b^*$ values (yellowness) by using a precise color reader (Shenzhen Wave Optoelectronics Technology Co).

## Sensory analysis

Sensory analysis was conducted at the end of the storage and measured with a SATAKE STA1B (No. 42570149, SATAKE Corporation, Japan) for the rice appearance, taste, and overall acceptability. A SATAKE RHS1A (No 41930183, SATAKE corporation, Japan) was used to determine rice hardness, elasticity, and stickiness. The samples were placed (30 g) in an aluminum container, washed with water until turbidity ceased the water, and then left to soak. The total time for washing and soaking was 30 min. Water with a final mass of 42 g was poured into the aluminum container. The samples were then placed in an electronic steam cooker and left to steam for 30 min, prior to cooling for 20 min. The cooled rice was placed in a circular steel ring of about 4 cm diameter and 1 cm width and compacted before measurement by using SATAKE STA1B and SATAKE RHS1A.

## Data analysis

Water content and RVA data are reported as the mean ± standard deviation (SD). Mean, SD, and ANOVA were computed using SPSS version 20.0. Data were analyzed using the least significance test, with graphing performed using Origin version 8.6. Correlation analysis was conducted using the pooled data of storage time, irradiation dose, and the following pasting properties: peak (PV), breakdown (BK), and setback (ST) viscosity, pasting temperature (PT), swelling power (Sp), color, and water content of rice flour and starch. Statistical significance was established at $p < 0.05$. Each sample was measured in triplicate.

## Results and discussion

### Water variance of white rice

White rice used for EBI contained 15.5% water at wet weight, 77.9% total starch, 5.75% protein, 0.355% total dietary fiber, 0.2% fat, and 0.28% ash. The variance of water content of native and irradiated white rice during storage is shown in Fig 1. The initial water content of white rice was 14.6%. During 75 days of storage, the water content of white rice at the storage temperature of 25 ˚C and 37 ˚C decreased to approximately 11.5% and 10.0%, respectively.

### Pasting properties of rice flour and starch

The major pasting properties of rice flour and starch were measured by RVA every 15 days within 75 days of storage at 25 ˚C and 37 ˚C (Tables 1–4). Selected pasting properties of rice flour and starch, including peak, breakdown, and setback viscosity, decreased in a dose-dependent manner. The extent of decrease in PV, BK, and ST values between adjacent two doses higher 2 kGy (Tables 1–4) was larger than the decrease between 0 and 2 kGy. Table 1 shows that the PV of native to irradiated rice flour at 8 kGy without storage declined considerably ($p < 0.05$) from 2327.3 cP to 308.5 cP. The increase in viscosity during the heating of the starch-containing suspension resulted from the swelling of starch granules, and the viscosity breakdown was due to the rupture of the swollen granules [24, 25]. Native rice flour and starch have high PV owing to the high granule and integrity because of the presence of amylose [26]. After EBI, a disruption in granule integrity and rigidity and the rupture of rice and starch granules occurred, thereby reducing the PV of rice flour and starch [26–28].

The BK of rice increased from approximately 600 cP (non-irradiated sample) to 800 cP at 2–4 kGy and decreased considerably to approximately 200 cP at 8 kGy before storage. By contrast, starch BK decreased from approximately 1000 cP at 0–4 kGy to 200 cP at 8 kGy. BK served as an indicator of granule fragility, the stability of flour and starch pastes and their resistance to shear force during heating. After irradiation, the surface of grain particles showed perforations, which gradually fell off to form a small, smooth, and spherical structures [12]. This result showed that rice starch granules can be destroyed by irradiation, and breakage intensified with increased dose [28]. These changes can inhibit the swelling of irradiated grains and

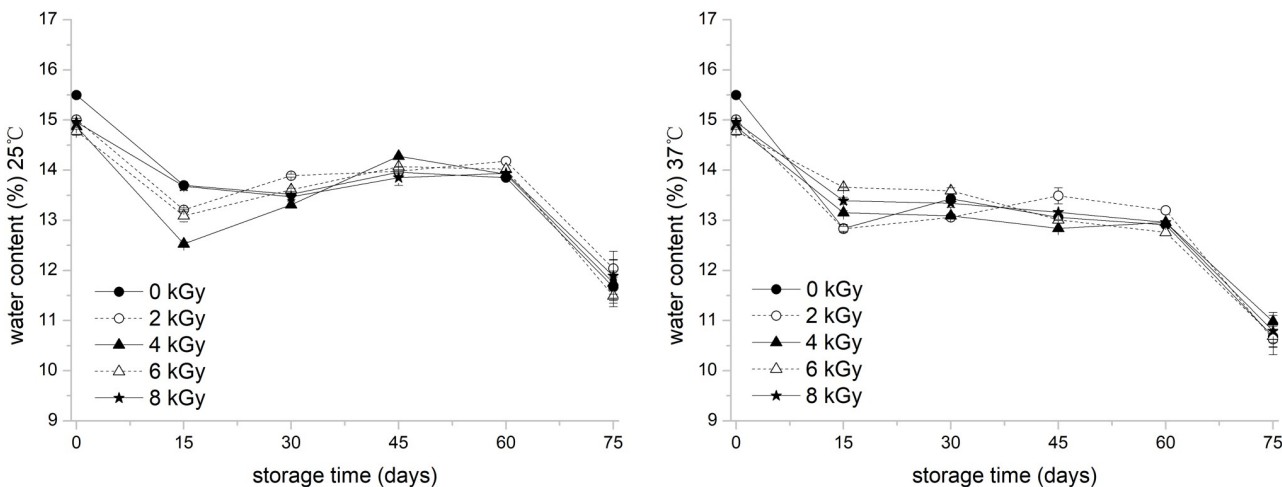

**Fig 1. Water content variance of electronic -beam -irradiated white rice at 25 ˚C and 37 ˚C with 75 days of storage.** Error bar represents standard deviation ($n = 3$).

**Table 1. Pasting properties of irradiated white rice flour during 75 days of storage at 25 ˚C.**

| Pasting properties | Treatment (kGy) | 0 | 15 | 30 | 45 | 60 | 75 |
|---|---|---|---|---|---|---|---|
| peak viscosity | 0 | 2327.3±23.0a(4) | 2455.3±45.6a(3) | 2528.5±24.7a(2,3) | 2517.5±24.7a(2,3) | 2743.5±46.0a(1) | 2557.0±32.5a(2) |
| | 2 | 2062.0±101.8b(1,2) | 2023.3±32.9b(2) | 2014.8±49.2b(2) | 2096.5±9.2b(1,2) | 2188.0±25.5b(1) | 2093±31.2b(1,2) |
| | 4 | 1302.5±20.5c(2) | 1244.3±32.9c(2,3) | 1256.8±33.6c(2,3) | 1194.5±24.7c(3) | 1374.0±25.5c(1) | 1294.5±24.7c(2) |
| | 6 | 630.8±1.1d(1) | 580.8±20.9d(2) | 575.5±21.9d(2) | 569.5±20.5d(2,3) | 553.0±12.7d(2) | 518.5±26.2d(3) |
| | 8 | 308.5±29.0e(1) | 239.5±23.3e(2) | 240.3±25.8e(2) | 223.0±19.8e(2) | 219.0±12.7e(2) | 215.0±7.1e(2) |
| breakdown viscosity | 0 | 612.5±17.7b(4) | 736.3±20.2b(3) | 715.5±21.9b(3) | 688.5±16.3c(3) | 1121.0±29.7b(1) | 963.5±27.6a(2) |
| | 2 | 883.8±25.1a(2,3) | 830.3±27.2a(3) | 851.5±26.2a(2,3) | 905.8±22.3a(2) | 1186.5±23.3a(1) | 852.0±25.5b(2,3) |
| | 4 | 867.0±21.2a(2) | 840.5±26.2a(2) | 842.3±28.6a(2) | 811.0±15.6b(2) | 1007.0±24.0c(1) | 944.5±34.6a(1) |
| | 6 | 473.8±33.6c(1) | 439.5±24.7c(1,2) | 435.0±18.4c(1,2) | 434.5±17.7d(1,2) | 415.0±21.2d(1,2) | 401.0±18.4c(2) |
| | 8 | 240.0±25.5d(1) | 182.3±17.3d(2) | 181.3±15.9d(2) | 169.0±12.7e(2) | 151.0±15.6e(2) | 151.5±16.3d(2) |
| setback viscosity | 0 | 1009.5±41.7a(2) | 1095.8±36.4a(2) | 1067.0±38.2a(2) | 1053.8±33.6a(2) | 1381.0±25.5a(1) | 1236.0±35.4a(1) |
| | 2 | 967.5±38.9a(2) | 914.8±20.9b(2) | 943.8±33.6b(2) | 963.5±33.2b(2) | 1122.0±31.1b(1) | 931.0±28.3b(2) |
| | 4 | 515.5±21.9b(1) | 478.3±25.8c(1,2) | 480.5±14.8c(1,2) | 452.0±31.1c(2) | 495.0±21.2c(1,2) | 460.0±14.1c(1,2) |
| | 6 | 176.0±8.5c(1) | 178.3±7.4d(1) | 185.8±8.1d(1) | 176.5±9.2d(1) | 173.0±4.2d(1) | 152.5±4.9d(2) |
| | 8 | 88.8±2.5d(1) | 74.0±1.4e(3) | 82.5±3.5e(2) | 72.0±2.8e(3) | 71.0±1.4e(3) | 61.5±2.1e(4) |
| pasting temperature | 0 | 70.6±0.5c(2,3) | 71.2±0.3c(2,3) | 72.3±0.4c(1) | 71.4±0.6c(1,2) | 71.1±0.2cd(2,3) | 70.3±0.2d(3) |
| | 2 | 70.7±0.3c(3,4) | 70.7±0.3d(3,4) | 72.4±0.3c(1) | 71.4±0.6c(2,3) | 70.4±0.2d(4) | 71.7±0.2c(1,2) |
| | 4 | 71.2±0.3c(2) | 72.3±0.3c(1) | 72.3±0.5c(1) | 72.3±0.4c(1) | 71.5±0.5c(1,2) | 72.3±0.4c(1) |
| | 6 | 84.5±0.4b(4) | 86.2±0.3b(3) | 86.4±0.5b(2,3) | 87.1±0.2b(1,2) | 87.1±0.1b(1,2) | 87.4±0.3b(1) |
| | 8 | 90.3±0.4a(2) | 90.4±0.5a(2) | 92.2±0.3a(1) | 91.3±0.4a(1,2) | 92.4±0.6d(2,3) | 92.4±0.6a(1) |

Values are means ± SD of three determinations ($n$ = 3).

Numbers following the lowercased letters mean significant differences.

Different numbers in a row or a column indicate significant differences at $p < 0.05$.

cause the decrease in BK, leading to shear-thinning behavior [29]. Conversely, cooked white rice irradiated at above 4 kGy may exhibit low resistance to oral chewing, thereby producing an undesired taste.

The STs of rice and starch at 0 kGy without storage was 1009.5 and 1357.8 cP, respectively, which decreased to 88.8 and 28.5 cP with increased irradiation dosage to 8 kGy. ST represented the degree of retrogradation of starch and primarily amylose and was significantly correlated with the degree of polymerization [24]. With increased EBI dose, the breakage of starch chains increased, especially the degradation of amylose and the long, linear amylopectin in the amorphous region containing primarily amylose; consequently, the degree of polymerization decreased [26, 30]. Therefore, ST was reduced in a dose-dependent manner.

The pasting temperature (PT) of rice flour stored at 25 ˚C and 37 ˚C was approximately 72 ˚C at 0–4 kGy and increased to 90 ˚C at 8 kGy with increased irradiation dose without storage. PT of starch flour decreased from 73.3 ˚C at 0 kGy to 71.4 ˚C at 8 kGy at 0 day. This difference would be further discussed in the last paragraph of this section. The irradiated samples stored at 37 ˚C exhibited higher PT than samples stored at 25 ˚C (Tables 1 and 2). This difference became distinct at 4 kGy and higher irradiation dose during storage. The low PT of rice flour and starch indicated their tendency to swell. The high PT value indicated that a considerable amount of water had to be absorbed during pasting [31]. The cooked rice samples with excessively high PT (irradiation dose > 4 kGy) were likely to be excessively soft and sticky. Therefore, 2 and 4 kGy were satisfactory irradiation doses for improving the cooking quality of white rice and a storage temperature of 25 ˚C for the improved rice quality.

**Table 2. Pasting properties of irradiated white rice flour during 75 days of storage at 37 ˚C.**

| Pasting properties | Treatment (kGy) | 0 | 15 | 30 | 45 | 60 | 75 |
|---|---|---|---|---|---|---|---|
| peak viscosity | 0 | 2327.3±23.0a(4) | 2321.0±29.7a(4) | 2706.8±57.6a(3) | 2815.3±54.1a(2) | 3019.0±43.8a(1) | 2953.0±46.7a(1) |
| | 2 | 2062.0±101.8b(3) | 2380.3±28.6a(1) | 2152.5±43.1b(2,3) | 2197.0±24.0b(2) | 2328.0±39.6b(1) | 2189.5±27.6b(2,3) |
| | 4 | 1302.5±20.5c(1) | 1237.5±23.3b(2) | 1084.8±20.9c(3) | 1048.5±37.5c(3) | 1084.0±24.0c(3) | 940.0±28.3c(4) |
| | 6 | 630.8±1.1d(1) | 510.0±14.1c(2) | 346.8±19.4d(3) | 311.5±16.3d(3,4) | 298.0±14.1d(4) | 256.0±17.0d(5) |
| | 8 | 308.5±29.0e(1) | 321.5±14.8d(1) | 136.0±7.1e(2) | 122.5±3.5e(2) | 135.5±6.4e(2) | 114.0±4.2e(2) |
| breakdown viscosity | 0 | 612.5±17.7b(6) | 900.5±14.8b(4) | 825.0±19.8b(5) | 970.8±15.2b(3) | 1471.5±30.4a(1) | 1360.5±29.0a(2) |
| | 2 | 883.8±25.1a(5) | 952.5±27.6a(4,5) | 994.5±20.5a(3,4) | 1061.5±44.5a(2,3) | 1127.5±24.7b(2) | 1201.5±19.1b(1) |
| | 4 | 867.0±21.2a(1) | 840.5±26.2c(1) | 722.5±16.3c(2) | 710.0±14.1c(2) | 755.0±18.4c(2) | 636.5±20.5c(3) |
| | 6 | 473.8±33.6c(1) | 380.3±14.5d(2) | 241.3±15.9d(3) | 227.0±8.5d(3) | 208.5±12.0d(3,4) | 175.5±12.0d(4) |
| | 8 | 240.0±25.5d(1) | 242.0±4.2e(1) | 84.0±5.7e(2) | 76.5±2.1e(2) | 82.0±2.8e(2) | 63.0±4.2e(2) |
| setback viscosity | 0 | 1009.5±41.7a(5) | 1107.0±26.9a(4) | 1232.3±30.1a(3) | 1319.8±27.9a(2) | 1476.5±30.4a(1) | 1454.5±34.6a(1) |
| | 2 | 967.5±38.9a(4) | 1107.0±38.2a(2,3) | 1058.0±39.6b(3) | 1124.5±34.6b(1,2,3) | 1201.5±30.4b(1) | 1191.5±16.3b(1,2) |
| | 4 | 515.5±21.9b(1) | 460.8±15.2b(2) | 444.5±16.3c(2,3) | 445.0±17.0c(2,3) | 474.0±11.3c(2) | 406.5±9.2c(3) |
| | 6 | 176.0±8.5c(1) | 174.3±6.0c(1) | 155.8±8.1d(2) | 141.5±4.9d(2,3) | 142.0±5.7d(2,3) | 124.5±9.2d(3) |
| | 8 | 88.8±2.5d(2) | 114.0±5.7d(1) | 62.0±2.8e(3) | 64.0±4.2e(3) | 64.5±3.5e(3) | 51.0±2.8e(4) |
| pasting temperature | 0 | 70.6±0.5c(3) | 71.3±0.4d(2,3) | 72.8±0.4c(1) | 72.3±0.4e(1,2) | 71.4±0.6d(2,3) | 73.0±0.7c(1) |
| | 2 | 70.7±0.3c(4) | 71.2±0.3d(4) | 73.4±0.6c(2) | 74.1±0.2d(1,2) | 74.3±0.2c(1) | 72.2±0.3c(3) |
| | 4 | 71.2±0.3c(4) | 74.5±0.4c(3) | 84.4±0.5b(2) | 85.3±0.4c(2) | 84.4±0.6b(2) | 87.1±0.2b(1) |
| | 6 | 84.5±0.4b(4) | 87.3±0.4b(3) | 91.2±0.2a(2) | 92.3±0.4b(1,2) | 91.6±0.5a(1,,2) | 92.5±0.6a(1) |
| | 8 | 90.3±0.4a(4) | 91.3±0.5a(3,4) | 92.0±0.0a(2,3) | 93.3±0.4a(1,2) | 92.0±0.7a(2,3) | 93.5±0.7a(1) |

Values are means ± SD of three determinations ($n$ = 3).

Numbers following the lowercased letters mean significant differences.

Different numbers in a row or a column indicate significant differences at $p < 0.05$.

The discrepancy of pasting properties of rice flour and starch at the same storage temperature in Tables 1–4 was partly due to the more extensive presence of protein, lipid, and sugar in rice than in alkali-isolated starch. For the starch samples, the pasting viscosities were higher than in the corresponding rice flour on average. This finding can be due to the fact that compared with the starch samples, rice flour had a higher protein content that prevented the penetration of water into the starch matrix upon denaturation [32]. The amylose–lipid complex formed in rice during heating resulted in entanglements in amylopectin molecules, which restricted the swelling of granules and caused the high PT and low PV [33]. The presence of sugars can increase PT because the water-binding ability of sugars decreases the availability for starch gelatinization [34, 35]. The discrepancy in pasting viscosity between rice and the corresponding starch subjected to EBI showed a dose-dependent decrease, as shown in Tables 1–4.

## Swelling power of the irradiated rice flour

RVA results showed that the pasting temperature of rice flour and starch was mostly between 70 ˚C and 90 ˚C. Therefore, the swelling power of rice flour and starch at 70 ˚C (Sp70) and 90 ˚C (Sp90) was analyzed because it can indicate the capacity of water absorption and swelling of the starch granules. Fig 2 shows that the swelling power of rice flour before storage decreased with increased irradiation dose and it was consistent with the results of Atrous et al. [36] and Liu et al. [13]. The mechanism of restricted swelling power of irradiated rice was similar to that of the PV reduction. EBI caused the rupture of rice and starch granules; as such, the granules exhibited less-complete structure and water-absorption ability that obstructed their

Table 3. Pasting properties of irradiated white rice starch during 75 days of storage at 25 ˚C.

| Pasting properties | Treatment (kGy) | 0 | 15 | 30 | 45 | 60 | 75 |
|---|---|---|---|---|---|---|---|
| peak viscosity | 0 | 2971.5±58.7a(1) | 3006.5±51.6a(1) | 2905.3±35.7a(1) | 2936.5±51.6a(1) | 2628.5±40.3a(2) | 2511.5±30.4a(3) |
| | 2 | 2274.5±48.8b(1) | 1962.5±46.0b(3) | 2106.8±37.8b(2) | 2253.0±46.7b(1) | 1724.0±33.9b(4) | 2065.0±49.5b(2,3) |
| | 4 | 1346.0±50.9c(1) | 1048.0±39.6c(4) | 1217.0±24.0c(2,3) | 1288.5±26.2c(1,2) | 1313.0±18.4c(1) | 1177.0±15.6c(3) |
| | 6 | 531.5±16.3d(4) | 570.3±14.5d(2,3) | 616.3±23.0d(1,2) | 645.0±21.2d(1) | 618.5±26.2d(1,2) | 617.5±24.7d(1,2) |
| | 8 | 240.5±4.9e(2) | 308.3±11.7d(1) | 248.3±11.7d(2) | 223.5±17.7d(2) | 246.0±18.4d(2) | 242.0±12.7d(2) |
| breakdown viscosity | 0 | 1084.8±35.0a(2) | 508.8±15.9b(4) | 507.0±18.4c(4) | 1048.0±39.6ab(2) | 950.5±43.1a(3) | 1324.5±62.9a(1) |
| | 2 | 953.3±47.0b(3) | 752.3±45.6a(4) | 725.3±35.7b(4) | 1122.5±31.8a(2) | 738.5±23.3b(4) | 1335.0±49.5a(1) |
| | 4 | 991.0±29.7b(1) | 711.3±15.9a(3) | 829.0±25.5a(2) | 973.0±32.5b(1) | 1003.5±33.2a(1) | 976.0±36.8b(1) |
| | 6 | 455.5±21.9c(2) | 466.3±23.0b(2) | 526.8±22.3c(1) | 565.0±28.3c(1) | 546.5±19.1c(1) | 549.0±22.6c(1) |
| | 8 | 201.0±15.6d(2) | 267.8±11.0c(1) | 208.3±11.7d(2) | 188.5±12.0d(2) | 208.5±12.0d(2) | 205.5±7.8d(2) |
| setback viscosity | 0 | 1357.8±53.4a(1) | 1233.3±47.0a(2) | 1183.8±33.6a(2,3) | 1428.5±40.3a(1) | 1433.5±47.4a(1) | 1083.0±32.5a(3) |
| | 2 | 611.3±15.9b(4) | 998.3±40.0b(2) | 690.3±18.7b(3) | 1057.0±49.5b(1,2) | 733.5±16.3b(3) | 1089.0±32.5a(1) |
| | 4 | 235.0±7.1c(4) | 412.8±18.0c(1) | 279.8±13.8c(3) | 342.5±13.4c(2) | 324.0±18.4c(2) | 399.5±16.3b(1) |
| | 6 | 75.0±14.1d(4) | 148.3±11.7d(1) | 100.0±8.5d(3) | 132.0±8.5d(1,2) | 122.5±6.4d(2,3) | 129.5±6.4c(1,2) |
| | 8 | 28.5±2.1d(4) | 78.5±3.5d(1) | 52.0±2.8d(3) | 62.5±4.9d(2) | 48.5±3.5e(3) | 53.0±4.2d(3) |
| pasting temperature | 0 | 73.3±0.3a(2) | 75.0±0.0b(1) | 75.3±0.3a(1) | 72.0±0.2ab(3) | 73.2±0.2b(2) | 70.3±0.5b(4) |
| | 2 | 72.1±0.2b(3) | 76.5±0.7a(1) | 74.9±0.2a(2) | 72.3±0.4ab(3) | 75.0±0.0a(2) | 71.5±0.4ab(3) |
| | 4 | 71.5±0.5b(2,3) | 74.3±0.4b(1) | 72.3±0.4c(2,3) | 71.4±0.6bc(2,3) | 71.3±0.4c(3) | 72.6±0.5a(2) |
| | 6 | 71.4±0.5b(1,2) | 72.3±0.4c(1) | 72.5±0.4c(1) | 70.6±0.6c(2) | 70.6±0.5c(2) | 68.4±0.9c(3) |
| | 8 | 71.4±0.6b(2,3) | 71.4±0.5c(2,3) | 72.3±0.4a(2) | 72.8±0.4a(1) | 72.3±0.4b(1,2) | 70.6±0.6b(3) |

Values are means ± SD of three determinations ($n$ = 3).

Numbers following the lowercased letters mean significant differences.

Different numbers in a row or a column indicate significant differences at $p < 0.05$.

swelling abilities [37]. The reduction in swelling power can also be explained by the ability of EBI to degrade the amylopectin chains, the gelatinized starch inhibited the diffusion of water into the starch matrix during heating [32]. The swelling power of irradiated rice stored at 25 ˚C and 37 ˚C and at 70 ˚C and 90 ˚C decreased after 75 days of storage. Furthermore, the swelling power of rice flour at each water bath temperature decreased more between adjacent doses at the storage temperature of 37 ˚C. The swelling power of starch depends on the water-holding capacity of starch molecules due to hydrogen bonding. During gelatinization, the hydrogen bonds that can stabilize the double helices in the crystallites are broken and replaced by hydrogen bonds with water [38]. From previous results of water variation during storage, the water content of rice flour stored at 25 ˚C was usually higher than at 37 ˚C, leading to increased hydrogen bonds among starch molecules, stable crystallite structure, and swelling.

## Rice color

With increased irradiation dose at 25 ˚C and 37 ˚C, L* (lightness) and b* (yellowness) values of the irradiated rice flour decreased and increased (in Fig 3), respectively, and were highly evident for irradiation doses above 4 kGy. These trends in color change may be due to the decomposition of glycosidic and peptide bonds during irradiation. In brief, the free radicals, including hydroxyl and hydrogen atoms, generated during irradiation can attack glycosidic bonds in carbohydrates and lead to depolymerization (degradation) and Maillard reaction [14]. The cleavage of the glycosidic bonds subsequently causes the starch granules to rupture, accelerating browning [15]. Moreover, the caramelization reaction of monosaccharides via the

**Table 4. Pasting properties of irradiated white rice starch during 75 days of storage at 37 ˚C.**

| Pasting properties | Treatment (kGy) | 0 | 15 | 30 | 45 | 60 | 75 |
|---|---|---|---|---|---|---|---|
| peak viscosity | 0 | 2971.5±58.7a(1) | 2923.5±47.4a(1) | 2967.5±53.0a(1) | 2861.0±43.8a(1) | 2478.0±39.6a(3) | 2649.0±41.0a(2) |
| | 2 | 2274.5±48.8b(2) | 2220.0±28.3b(2) | 2208.3±40.0b(2) | 2295.5±21.9b(2) | 2111.5±30.4b(3) | 3468.0±39.6b(1) |
| | 4 | 1346.0±50.9c(1) | 1016.0±28.3c(3) | 1259.8±22.3c(2) | 1227.5±23.3c(2) | 1250.5±24.7c(2) | 1195.0±21.2c(2) |
| | 6 | 531.5±16.3d(2) | 424.5±19.1d(3) | 620.3±14.5d(1) | 623.0±18.4d(1) | 618.5±12.0d(1) | 567.5±10.6d(2) |
| | 8 | 240.5±4.9e(2) | 285.0±7.1e(1) | 230.0±7.1e(2,3) | 227.5±7.8e(2,3) | 239.5±9.2e(2) | 216.5±9.2e(3) |
| breakdown viscosity | 0 | 1084.8±35.0a(2) | 458.0±19.8c(5) | 515.3±21.6b(5) | 713.5±33.2c(4) | 817.5±24.7c(3) | 1231.5±44.5b(1) |
| | 2 | 953.3±47.0b(4) | 769.5±27.6a(5) | 895.0±21.2a(4) | 1276.5±37.5a(2) | 1151.0±45.3a(3) | 2991.0±58.0a(1) |
| | 4 | 991.0±29.7b(1) | 536.0±19.8b(2) | 922.0±31.1a(1) | 947.5±38.9b(1) | 969.5±55.9b(1) | 936.0±22.6c(1) |
| | 6 | 455.5±21.9c(3) | 282.5±17.7d(4) | 530.5±14.8b(1,2) | 550.0±15.6d(1) | 543.0±14.1d(1) | 490.5±14.8d(2,3) |
| | 8 | 201.0±15.6d(1) | 200.5±6.4e(1) | 194.5±6.4c(1,2) | 191.5±2.1e(1,2) | 195.5±7.8e(1,2) | 174.5±6.4e(2) |
| setback viscosity | 0 | 1357.8±53.4a(1) | 1081.8±44.9a(2,3) | 1191.8±44.9a(2) | 1430.5±43.1a(1) | 1433.0±46.7a(1) | 1044.0±46.7a(3) |
| | 2 | 611.3±15.9b(4) | 658.0±35.4b(3,4) | 731.8±36.4b(3) | 1163.0±26.9b(1) | 954.5±30.4b(2) | 534.5±20.5b(5) |
| | 4 | 235.0±7.1c(4) | 348.5±21.9c(2) | 288.3±25.8c(3) | 424.5±19.1c(1) | 382.0±22.6c(1,2) | 380.5±14.8c(1,2) |
| | 6 | 75.0±14.1d(3) | 141.5±12.0d(1) | 106.0±8.5d(2) | 153.0±7.1d(1) | 127.0±14.1d(1,2) | 150.5±9.2d(1) |
| | 8 | 28.5±2.1d(5) | 113.0±4.2d(1) | 46.5±2.1d(4) | 57.5±3.5d(2,3) | 64.5±4.9d(2) | 50.0±4.2d(3,4) |
| pasting temperature | 0 | 73.3±0.3a(4) | 76.1±0.2a(1) | 75.3±0.4a(2) | 74.5±0.4a(2,3) | 75.3±0.3a(2) | 74.2±0.3a(3) |
| | 2 | 72.1±0.2b(2) | 74.6±0.3b(1) | 73.5±0.6b(1) | 71.4±0.5b(2) | 71.4±0.5c(2) | 68.3±0.4c(3) |
| | 4 | 71.5±0.5b(1) | 71.3±0.4c(1) | 71.4±0.6c(1) | 71.4±0.6b(1) | 70.6±0.5c(1) | 71.3±0.4b(1) |
| | 6 | 71.4±0.5b(1,2) | 71.3±0.4c(1,2) | 72.1±0.1c(1) | 70.7±0.5b(2,3) | 70.1±0.1c(3) | 70.5±0.7b(2,3) |
| | 8 | 71.4±0.6b(2) | 73.8±0.4b(1) | 72.4±0.5bc(1,2) | 74.0±0.6a(1) | 73.8±0.8b(1) | 73.8±0.8a(1) |

Values are means ± SD of three determinations ($n = 3$).

Numbers following the lowercased letters mean significant differences.

Different numbers in a row or a column indicate significant differences at $p < 0.05$.

degradation of starch polysaccharide during irradiation can be another explanation for the color change, as described by Sc et al. [39].

The L* value of the samples decreased within the first month and then increased to approximately the initial L* value within the remaining storage time. The b* value of samples irradiated at 4 kGy and above increased notably after storage for half a month, whereas the b* value slightly increased at irradiation doses of 0 and 2 kGy. The magnitude of the decreased b* and L* values among different irradiation dose of samples stored at 37 ˚C was more pronounced than those at 25 ˚C. Considering that rice with high brightness and low brown values is favorable to consumers, low irradiation dose (less than 4 kGy) with room temperature storage is the preferred treatment for commercial applications.

## Correlations among the properties of irradiated rice and starch

Pearson's correlation coefficients for the relationship among different functional properties of rice subjected to EBI during storage at 25 ˚C and 37 ˚C are shown in Tables 5 and 6, respectively. The rice storage time at 25 ˚C exhibited a negative correlation with Sp70 ($r = -0.672$, $p < 0.01$), Sp90 ($r = -0.739$, $p < 0.01$), and water content ($r = -0.654$, $p < 0.01$), whereas the correlation coefficients were −0.693 ($p < 0.01$), −0.747 ($p < 0.01$), and −0.859 ($p < 0.01$) at 37 ˚C. Storage time was negatively correlated with L* value (r = −0.386, $p < 0.05$) at the storage temperature of 37 ˚C. No other significant relationships were found between storage time and the functional properties of irradiated rice and starch. This result differed from that for long-

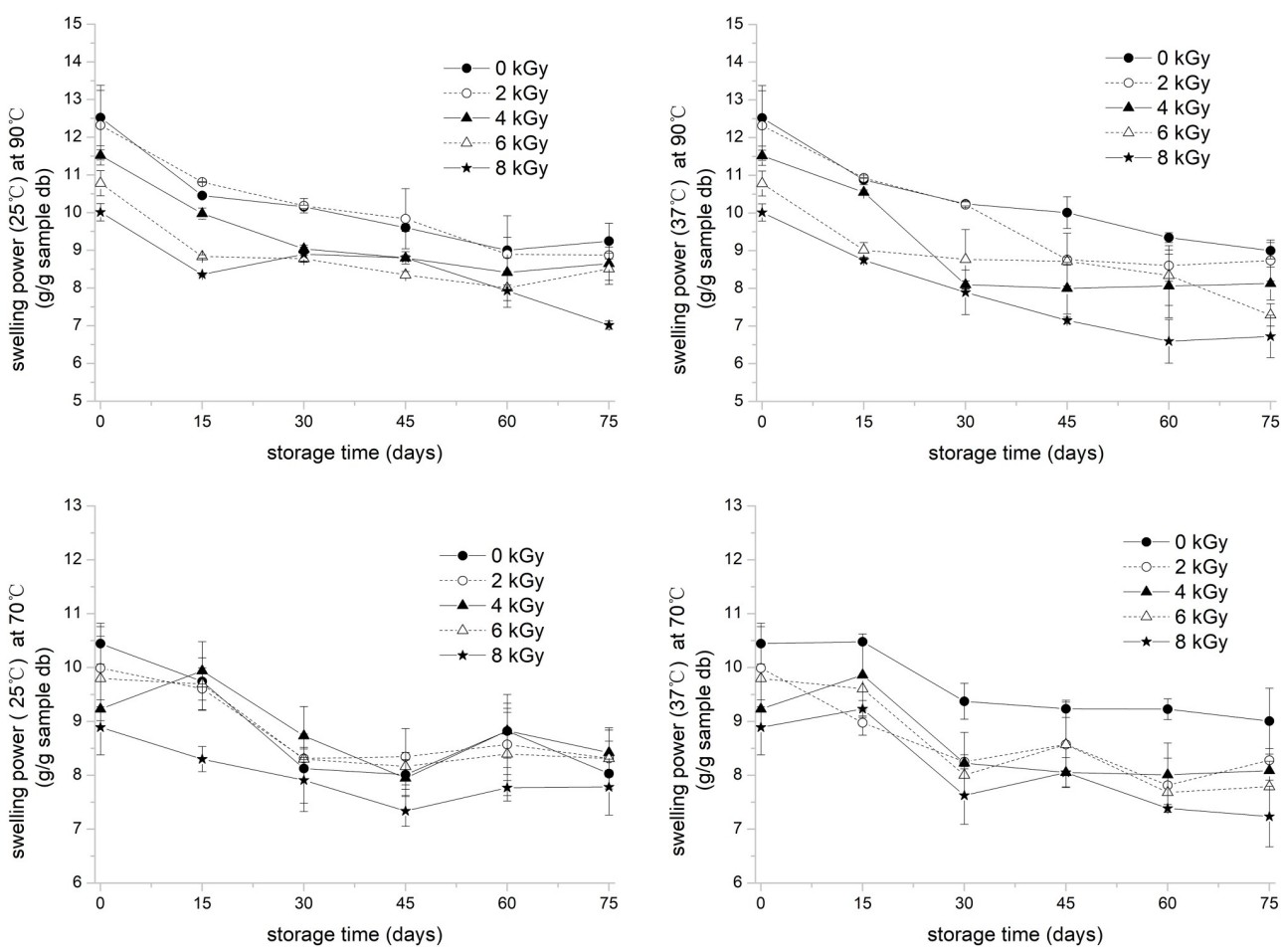

**Fig 2. Swelling power changes at 70 ˚C and 90 ˚C of the electronic -beam -irradiated white rice stored at 25 ˚C and 37 ˚C for 75 days of storage.** Error bar represents standard deviation (*n* = 3).

term storage of rice reported by Chen et al. [15], showing that storage time had a negative relationship with ST and PT at a significance level of $p < 0.01$. The discrepancy may be due to disparate sources of irradiation, storage conditions, and rice variants, i.e., storage significantly decreased PT at 15 months (not significant within 6 months) and may have been due to rice aging, which can form disulfide bonds among proteins that accumulate with prolonged time [40]. The lipid and protein content were relatively higher in brown rice than in white rice. During storage, lipid is hydrolyzed by lipase into free fatty acid, thereby accelerating rice aging and PT reduction [15]. Starch is fragmented into short linear polymers after EBI or gamma irradiation and may be unable to form a helical structure with fatty acids during storage; as such, the passage of water may be restricted, and thus decrease the pasting viscosities with prolonged storage [16, 41]. The damage to starch at the same dose can be worsened by gamma irradiation, which has a high penetration power. A significant negative correlation between storage time and PT or ST was not observed because the duration of storage in the current study was not sufficiently long. The rice water content showed significant correlation with Sp70 and Sp90 at 37 ˚C, and the correlation coefficients were 0.625 ($p < 0.01$) and 0.515 ($p < 0.01$), respectively. At 25 ˚C storage, a positive correlation with only Sp90 (r = 0.54323, $p < 0.01$) was observed.

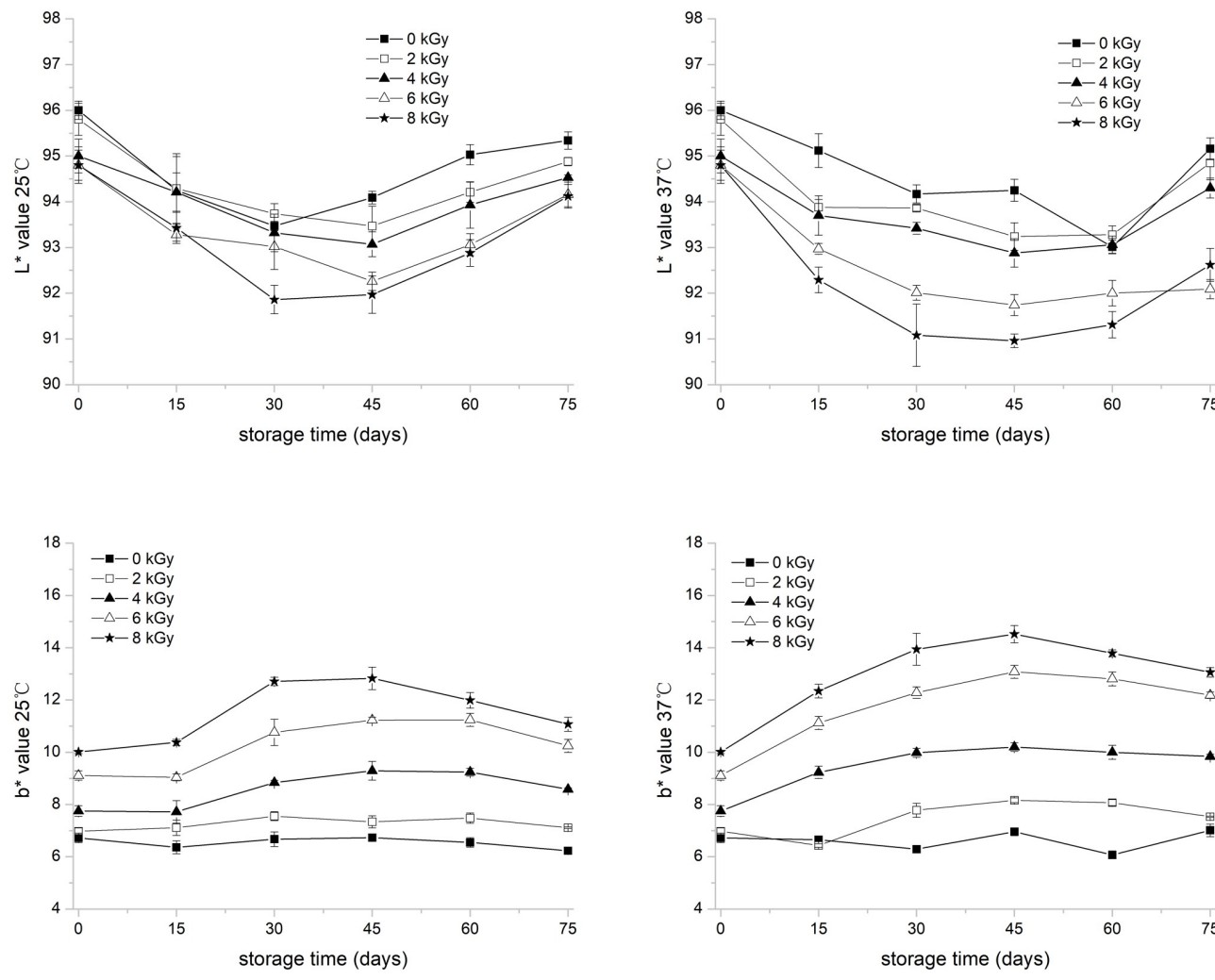

**Fig 3. Brightness (L\*) and yellowness value (b\*) changes of electronic -beam -irradiated white rice at 25 ˚C and 37 ˚C with 75 days of storage.** Error bar represents standard deviation (*n* = 3).

Conversely, increased irradiation dose led to negative correlations with nearly all functional properties (Tables 5 and 6) except for PT-F and b\*. The correlation coefficients between PT-F and irradiation dose at storage temperature of 25 ˚C and 37 ˚C were 0.902 ($p < 0.01$) and 0.909 ($p < 0.01$), respectively. The positive correlation coefficients between b\* and irradiation dose were 0.926 ($p < 0.01$) and 0.913 ($p < 0.01$) for the storage temperatures of 25 ˚C and 37 ˚C, respectively.

## Sensory qualities of cooked rice

In accordance with the instruction manual of SATAKE RHS1A and STA1B, the standard values of rice stickiness, elasticity, and hardness ranged between 0.5–0.6, 0.7–0.72, and 4–5, respectively. No standard scores were given for appearance, taste, and overall acceptability, with high values, corresponding to good rice quality. The full scores for the stickiness, elasticity, hardness, appearance, taste, and overall acceptability of the irradiated rice samples were 1, 1, 10, 10, 10, and 100, respectively. Fig 4 shows that the value of elasticity and hardness of the rice sample decreased with increased EBI dose. The value of elasticity and hardness of

Table 5. Correlation coefficients among various functional properties of the irradiated rice sample at 25 °C[a].

| | st | dose | PV-F | BK-F | ST-F | PT-F | PV-S | BK-S | ST-S | PT-S | Sp90 | Sp70 | L* | b* |
|---|---|---|---|---|---|---|---|---|---|---|---|---|---|---|
| dose | | | | | | | | | | | | | | |
| PV-F | 0.02 | -0.989** | | | | | | | | | | | | |
| BK-F | 0.118 | -0.800** | 0.812** | | | | | | | | | | | |
| ST-F | 0.043 | -0.966** | 0.991** | 0.799** | | | | | | | | | | |
| PT-F | 0.041 | 0.902** | -0.897** | -0.935** | -0.861** | | | | | | | | | |
| PV-S | -0.047 | -0.981** | 0.977** | 0.704** | 0.956** | -0.847** | | | | | | | | |
| BK-S | 0.198 | -0.754** | 0.739** | 0.816** | 0.705** | -0.844** | 0.692** | | | | | | | |
| ST-S | 0.047 | -0.936** | 0.946** | 0.626** | 0.939** | -0.757** | 0.958** | 0.641** | | | | | | |
| PT-S | -0.349 | -0.457* | 0.489** | 0.365* | 0.496** | -0.445* | 0.458* | 0.001 | 0.467** | | | | | |
| Sp90 | -0.739** | -0.521** | 0.504** | 0.341 | 0.478** | -0.527** | 0.553** | 0.340 | 0.438* | 0.401* | | | | |
| Sp70 | -0.672** | -0.306 | 0.270 | 0.324 | 0.241 | -0.384* | 0.268 | 0.190 | 0.190 | 0.318 | 0.695** | | | |
| L* | -0.176 | -0.555** | 0.560** | 0.498** | 0.555** | -0.560** | 0.537** | 0.556** | 0.495** | 0.034 | 0.596** | 0.583** | | |
| b* | 0.158 | 0.926** | -0.925** | -0.802** | -0.903** | 0.896** | -0.897** | -0.730** | -0.846** | -0.436* | -0.604** | -0.500** | -0.725** | |
| w | -0.654** | -0.048 | 0.054 | -0.017 | 0.056 | -0.057 | 0.111 | -0.069 | 0.024 | 0.195 | 0.543** | 0.311 | 0.069 | -0.031 |

[a] * and ** indicate significance at $p < 0.05$ and $p < 0.01$, respectively.

Abbreviations: F = flour, S = starch, st = storage time, PK = peak viscosity, BK = breaking viscosity, ST = setback viscosity, PT = pasting temperature, Sp90 = swelling power at 90 °C, Sp70 = swelling power at 70 °C, and w = water content.

rice stored at 37 °C was higher than that of rice stored at 25 °C. Cooked white rice irradiated at 2 and 4 kGy at each storage temperature achieved scores near the standard scores and performed better than other irradiated rice in terms of elasticity and hardness. White rice irradiated at above 4 kGy evidently exhibited a low score of hardness (approximately 3), consistent with the results of the swelling power testing. The stickiness value of samples irradiated at 8 kGy was much higher when stored at 37 °C that than upon storage at 25 °C. Moreover, rice

Table 6. Correlation coefficients among various functional properties of the irradiated rice sample at 37 °C[b].

| | st | dose | PV-F | BK-F | ST-F | PT-F | PV-S | BK-S | ST-S | PT-S | Sp90 | Sp70 | L* | b* |
|---|---|---|---|---|---|---|---|---|---|---|---|---|---|---|
| dose | | | | | | | | | | | | | | |
| PV-F | -0.005 | -0.966** | | | | | | | | | | | | |
| BK-F | 0.071 | -0.882** | 0.918** | | | | | | | | | | | |
| ST-F | 0.072 | -0.950** | 0.993** | 0.918** | | | | | | | | | | |
| PT-F | 0.232 | 0.909** | -0.933** | -0.891** | -0.908** | | | | | | | | | |
| PV-S | 0.029 | -0.955** | 0.950** | 0.847** | 0.953** | -0.900** | | | | | | | | |
| BK-S | 0.284 | -0.550** | 0.543** | 0.667** | 0.582** | -0.558** | 0.676** | | | | | | | |
| ST-S | 0.041 | -0.924** | 0.932** | 0.777** | 0.924** | -0.812** | 0.887** | 0.384* | | | | | | |
| PT-S | -0.153 | -0.329 | 0.400* | 0.165 | 0.376* | -0.285 | 0.277 | -0.406* | 0.485** | | | | | |
| Sp90 | -0.747** | -0.564** | 0.546** | 0.444* | 0.478** | -0.722** | 0.529** | 0.163 | 0.453* | 0.205 | | | | |
| Sp70 | -0.693** | -0.509** | 0.480** | 0.381* | 0.407* | -0.605** | 0.449* | 0.023 | 0.458* | 0.327 | 0.854** | | | |
| L* | -0.386* | -0.664** | 0.642** | 0.607** | 0.594** | -0.747** | 0.669** | 0.460* | 0.511** | 0.086 | 0.790** | 0.693** | | |
| b* | 0.234 | 0.913** | -0.926** | -0.885** | -0.893** | 0.947** | -0.884** | -0.522** | -0.812** | -0.285 | -0.731** | -0.632** | -0.833** | |
| w | -0.859** | 0.012 | -0.003 | -0.085 | -0.067 | -0.174 | -0.056 | -0.266 | 0.037 | 0.072 | 0.625** | 0.515** | 0.247 | -0.160 |

[b] * and ** indicate significance at $p < 0.05$ and $p < 0.01$, respectively.

Abbreviations: F = flour, S = starch, st = storage time, PK = peak viscosity, BK = breaking viscosity, ST = setback viscosity, PT = pasting temperature, Sp90 = swelling power at 90 °C, Sp70 = swelling power at 70 °C, and w = water content.

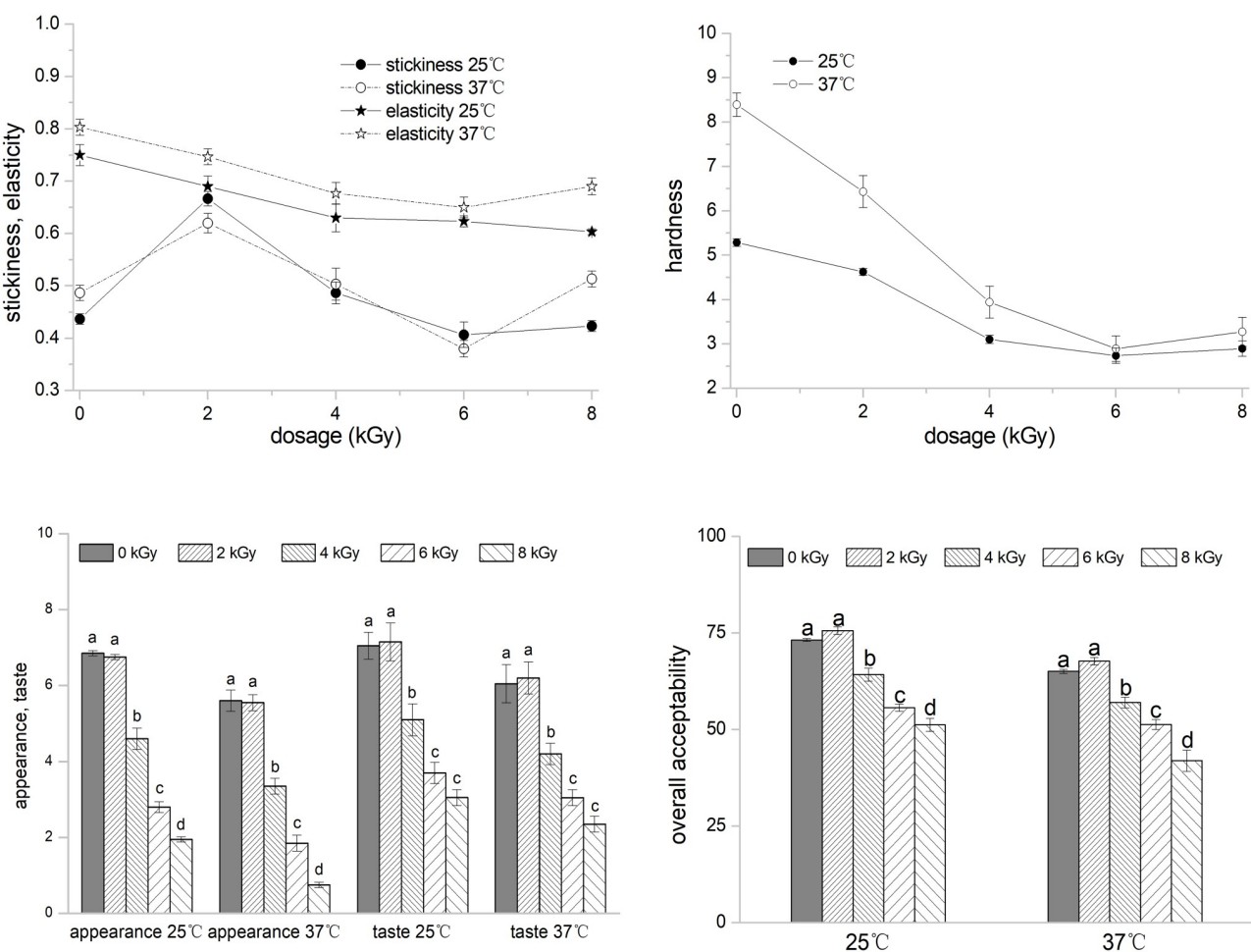

**Fig 4. Effects of EBI on the sensory qualities of cooked rice stored for 75 days at 25 ˚C and 37 ˚C.** Different letters/numbers above the bar indicate significant differences at $p < 0.05$. Error bar represents standard deviation ($n$ = 3).

irradiated at 2 kGy at each temperature exhibited a higher stickiness value (0.62–0.67) than rice irradiated at other doses (0.38–0.51). For taste, appearance, and overall acceptability, the scores decreased significantly ($p < 0.05$) with a high irradiation dose (4 kGy and above). In comparison, the sensory evaluation of white rice irradiated at 2 kGy were better compared with those of non-irradiated samples. Therefore, white rice with relatively low doses was the most acceptable.

## Conclusion

The physicochemical and sensory properties of irradiated white rice during short-term storage at 25 ˚C and 37 ˚C were characterized. At 25 ˚C and 37 ˚C, low or medium doses of EBI from 2 kGy to 4 kGy decreased the major pasting properties and improved the storage performance of white rice. Conversely, when white rice was irradiated at above 4 kGy, poor performance in color and sensory evaluation was observed at high storage temperatures. Therefore, EBI doses between 2 and 4 kGy were suitable for white rice treatment prior to short-term, low temperature storage. These results support the basis for developing an alternative method for white-rice storage.

## Supporting information

**S1 Table. Water content of electronic -beam -irradiated white rice at 25 ˚C and 37 ˚C with 75 days of storage.** Values are means ± SD of three determinations (n = 3). Numbers following the lowercased letters mean significant differences. Different numbers in a row or a column indicate significant differences at $p < 0.05$.
(PDF)

**S2 Table. Swelling power changes at 70 ˚C and 90 ˚C of the electronic -beam -irradiated white rice stored at 25 ˚C and 37 ˚C for 75 days of storage.** Values are means ± SD of three determinations (n = 3). Numbers following the lowercased letters mean significant differences. Different numbers in a row or a column indicate significant differences at $p < 0.05$.
(PDF)

**S3 Table. Brightness ($L^*$) and yellowness value ($b^*$) changes of electronic -beam -irradiated white rice at 25 ˚C and 37 ˚C with 75 days of storage.** Values are means ± SD of three determinations (n = 3). Numbers following the lowercased letters mean significant differences. Different numbers in a row or a column indicate significant differences at $p < 0.05$.
(PDF)

**S4 Table. Effects of EBI on the sensory qualities of cooked rice stored for 75 days at 25 ˚C and 37 ˚C.** Values are means ± SD of three determinations (n = 3). Different letters in a column indicate significant differences at $p < 0.05$.
(PDF)

## Author Contributions

**Conceptualization:** Zhihong Du, Jiali Xing, Lihong Pan, Yulin Li, Ren Wang, Yuntao Liu.

**Data curation:** Zhihong Du, Jiali Xing, Xiaohu Luo, Li Wang.

**Formal analysis:** Zhihong Du, Jiali Xing, Xiaohu Luo, Ren Wang, Yuntao Liu, Xiaohong Li.

**Investigation:** Zhihong Du.

**Methodology:** Zhihong Du, Jiali Xing, Lihong Pan, Yulin Li.

**Resources:** Xiaohu Luo, Li Wang, Zhengxing Chen.

**Software:** Zhihong Du, Ren Wang.

**Supervision:** Xiaohu Luo, Li Wang, Zhengxing Chen.

**Validation:** Zhihong Du, Xiaohong Li.

**Writing – original draft:** Zhihong Du.

**Writing – review & editing:** Zhihong Du, Jiali Xing.

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
