## [Decision Letter · Decision Letter 0]

23 Jul 2019

PONE-D-19-15367

Characterization of physical properties of electron-beam-irradiated white rice and starch during short-term storage

PLOS ONE

Dear Dr. Luo,

Thank you for submitting your manuscript to PLOS ONE. After careful consideration, we feel that it has merit but does not fully meet PLOS ONE’s publication criteria as it currently stands. Therefore, we invite you to submit a revised version of the manuscript that addresses the points raised during the review process.

We would appreciate receiving your revised manuscript by Sep 06 2019 11:59PM. To enhance the reproducibility of your results, we recommend that if applicable you deposit your laboratory protocols in protocols.io, where a protocol can be assigned its own identifier (DOI) such that it can be cited independently in the future. For instructions see: http://journals.plos.org/plosone/s/submission-guidelines#loc-laboratory-protocols

We look forward to receiving your revised manuscript.

Kind regards,

Walid Elfalleh, Ph.D

Academic Editor

PLOS ONE

Journal Requirements:

2. Please amend the manuscript submission data (via Edit Submission) to include author Zhengxing Chena.

Reviewers' comments:

Reviewer's Responses to Questions

**Comments to the Author**

1. Is the manuscript technically sound, and do the data support the conclusions?

Reviewer #1: Yes

Reviewer #2: Partly

2. Has the statistical analysis been performed appropriately and rigorously? 

Reviewer #1: Yes

Reviewer #2: Yes

3. Have the authors made all data underlying the findings in their manuscript fully available?

Reviewer #1: Yes

Reviewer #2: No

4. Is the manuscript presented in an intelligible fashion and written in standard English?

Reviewer #1: Yes

Reviewer #2: No

5. Review Comments to the Author

Reviewer #1: In this manuscript, authors investigate in the effect of electron-beam-irradiation on physical properties of white rice and starch during short-term storage. This is a highly important topic and there are many interesting results. However, there are some weak points relative to the method and sometimes the language should be improved throughout the manuscript. In the present form it is not possible to recommend publication for the paper. Yet, with a lot of additional work to improve the manuscript it may be possible to develop the paper into a publishable manuscript. Some other remarks are listed below:

Introduction : Explain well the topic but it is concentrated on irradiation effect on microorganisms which is not treated in this work.

A section of studying the effect on microorganisms should be added in this manuscript.

MM :

line 98-99 should be in the result

Line 96-97: you have to explain the choice of the dose spectrum

Irradiation, starch isolation, pasting proprieties: lack of reference

Results and discussion: well explained

Figures are not clear

Reviewer #2: PONE-D-19-15367

Comments to Authors-

-The authors extensively investigate the effect of electronic-beam irradiation on the storage quality of white rice at two different ambient temperatures. In general, the data collection is complete. However, there are some issues in the data presentation, result demonstration and discussion as well. Detailed comments are as follows:

Introduction-

-The difference between rice flour and starch should be better described in the introduction section. Indeed, this should be closely related to the results/discussion part where the effects of EBI on the rice quality properties are presented in Table1-4.

Materials and methods-

-L90-91 it is not clear that where these nutrient data come from. This should be better explained.

-L95- ‘a dose rate of 2kGy/h’ needs to be deleted because multiple doses of irradiation were applied, as stated in L96-97. Moreover, the duration of the irradiation treatment (which I think is critical) should be mentioned.

-L98- it was not described how the moisture content (at wet basis) was measured in the method section, and what the changes of this parameter would affect other properties such as Sp, rice color and storage time. In addition, the moisture content should also be included in the correlation analysis.

Results and discussion-

- Discussion is not sufficient (for example, L165-167, L216-218, L239-242, L284-285); it is suggested to refer to more relevant literatures and analyze each measure parameter on by one.

-The effect of EBI on the microbes should be mentioned and discussed somewhere as well.

Tables and Figures -

-In Table1-4, what do those numbers following the lower letters mean? It should be explained in the table footnote. In the footnote of all the four tables: ‘In a row, no numbers in common are significantly different (p < 0.05). In a column within one RVA pasting property, no letters in common are significantly different (p < 0.05)’. Consider revising - ‘different numbers in either a row or a column indicate significant differences at p < 0.05’. And the same footnote doesn’t need to be repeated four times.

-The font in Fig 1-3 looks small and unnormal. Consider revising.

- In Fig.3, the meaning of the letters/numbers above the bars should be explained in the figure legend.

Overall, the text should be carefully checked for English spelling and grammar (singular/plural, tenses, use of articles…); it is suggested to find a native English-speaking expert or a professional editing service to modify the whole manuscript. The literatures format should be carefully checked as well.

6. PLOS authors have the option to publish the peer review history of their article (what does this mean?). If published, this will include your full peer review and any attached files.

Reviewer #1: Yes: Hamza Hammadi

Reviewer #2: No

---

## [Author Response · Author response to Decision Letter 0]

15 Sep 2019

Response to Reviewer #1

Comments 1: Introduction: Explain well the topic but it is concentrated on irradiation effect on microorganisms which is not treated in this work.

A section of studying the effect on microorganisms should be added in this manuscript.

Response: Thank you for your kind suggestion. This work was based on the effect of electron-beam irradiation (EBI) on microorganisms. Thus, we discussed the topic in depth in the introduction. In our previous work entitled “Effect of electron beam irradiation on storability of brown rice and milled rice,” we found that the total viable bacterial count was significantly reduced from around 3×103 CFU/g at 0 kGy to 10 and 0 CFU/g at 5 kGy. Therefore, the present work did not anymore focus on the effects of EBI on microorganisms. The explanation for this point has been added in the introduction (seen in page 5, 6, line 86-90). We hope that you will be satisfied with our revisions. 

Comments 2: Line 98-99 should be in the result

Response: Thank you for this comment. The water content analysis was added in the results (seen in page 11, line 167-177; Fig 1).

Comments 3: Line 96-97: you have to explain the choice of the dose spectrum Irradiation, starch isolation, pasting proprieties: lack of reference

Response: The explanation on the choice of the dose spectrum irradiation (seen in page 6, line 101) and references of starch isolation (seen in page 7, line 112-113) and pasting properties (seen in page 8, line 126-127) has been added in the manuscript. 

Comments 4: Figures are not clear

Response: The figures have been modified (seen in Figs 1-4). Thank you very much.

Response to Reviewer #2

Comments 1: Introduction-

-The difference between rice flour and starch should be better described in the introduction section. Indeed, this should be closely related to the results/discussion part where the effects of EBI on the rice quality properties are presented in Table1-4.

Response: Thank you for your suggestion. The difference between rice flour and starch has been discussed in the introduction (seen in page 5, line 79-85), and the pasting properties related to it have also been presented (seen in page 19, 20, line 237-247).

Comments 2: Materials and methods-

-L90-91 it is not clear that where these nutrient data come from. This should be better explained.

Response: Thank you for highlighting this point. The determination methods and references of nutrient data have been added in the manuscript (seen in page 7, line 107-109).

Comments 3: -L95- ‘a dose rate of 2kGy/h’ needs to be deleted because multiple doses of irradiation were applied, as stated in L96-97. Moreover, the duration of the irradiation treatment (which I think is critical) should be mentioned.

Response: The part “a dose rate of 2 kGy/h” has been deleted (seen in page 6, line 100). Moreover, the irradiation duration was fixed because the item was placed on a speed-constant conveyor belt and passed through fixed distance. The variance during electron beam irradiation was irradiation intensity controlled by electronic speed flow that varied with the irradiation dose. The 5 s duration of irradiation has been added to the manuscript following your suggestion (seen in page 6, line 99).

Comments 4: -L98- it was not described how the moisture content (at wet basis) was measured in the method section, and what the changes of this parameter would affect other properties such as Sp, rice color and storage time. In addition, the moisture content should also be included in the correlation analysis.

Response: The methods of determining the moisture content of white rice and the corresponding results (including the figure legend) have been added in the manuscript (seen in page 7, line 107-108; page 11, line 167-177; and Fig 1). The moisture content has also been included in the correlation analysis (seen in Tables 5-6). In accordance with the correlation analysis, water content has a significant correlation with storage time and swelling power (seen in page 23, line 300-301; page 24, line 317-320). Therefore, the relationship between water and Sp was discussed (seen in page 21, line 262-267). 

Comments 5: Results and discussion-

- Discussion is not sufficient (for example, L165-167, L216-218, L239-242, L284-285); it is suggested to refer to more relevant literatures and analyze each measure parameter on by one.

Response: Thank you for your suggestion. We have analyzed each parameter individually, which we hope will now meet your standard (seen in page 12, line 186-191; seen in page 13, line 203-208; seen in page 20, line 254-259; seen in page 23, 24, line 307-317).

Comments 6: The effect of EBI on the microbes should be mentioned and discussed somewhere as well.

Response: Thank you for your kind suggestion. In our previous work entitled “Effect of electron beam irradiation on storability of brown rice and milled rice,” we found that total viable bacterial count was significantly reduced from approximately 3×103 CFU/g at 0 kGy to 10 and 0 CFU/g at 5 kGy (seen in page 5, 6, line 86-90). Therefore, the present work did not anymore focus on the effects of EBI on microorganisms. A relevant explanation has been added in the introduction.

Comments 7: Tables and Figures -

-In Table1-4, what do those numbers following the lower letters mean? It should be explained in the table footnote. In the footnote of all the four tables: ‘In a row, no numbers in common are significantly different (p < 0.05). In a column within one RVA pasting property, no letters in common are significantly different (p < 0.05)’. Consider revising - ‘different numbers in either a row or a column indicate significant differences at p < 0.05’. And the same footnote doesn’t need to be repeated four times.

Response: We appreciate your helpful suggestions. The numbers that follow the lowercased letters indicate significant differences. This point has been explained in the table footnote. The footnote was as follows: “In a row, no numbers in common are significantly different (p < 0.05). In a column within one RVA pasting property, no letters in common are significantly different (p < 0.05)” has been revised to “Different numbers in a row or a column indicate significant differences at p < 0.05.” The repetitive footnotes have been deleted (seen in page 18, line 224-225).

Comments 8: -The font in Fig 1-3 looks small and unnormal. Consider revising.

Response: We appreciate this comment. The font in Figs. 1–3 has been revised (seen in Figs. 2–4).

Comments 9: - In Fig.3, the meaning of the letters/numbers above the bars should be explained in the figure legend.

Response: Thank you for this reminder. The meaning of the letters above the bars has been explained in the figure legend (seen in page 28, line 356).

Comments 10: Overall, the text should be carefully checked for English spelling and grammar (singular/plural, tenses, use of articles…); it is suggested to find a native English-speaking expert or a professional editing service to modify the whole manuscript. The literatures format should be carefully checked as well.

Response: We appreciate your helpful suggestions. The entire manuscript has been rechecked by a professional editing service. The literature format has been carefully checked. We hope that the revised version would meet the standard of PLOS ONE.

We are grateful for your helpful comments.

References:

1. Goel PK, Singhal RS, Kulkarni PR. Studies on interactions of corn starch with casein and casein hydrolysates. Food Chemistry. 1999; 64(3): 383-9. doi: 10.1016/s0308-8146(98)00134-4. PubMed PMID: WOS: 000077881500014.

2. Luo XH, Li YL, Yang D, Xing JL, Li K, Yang M, et al. Effects of electron beam irradiation on storability of brown and milled rice. Journal of Stored Products Research. 2019; 81: 22-30. doi: 10.1016/j.jspr.2018.12.003. PubMed PMID: WOS: 000463305500004.

3. Chen X, Jin Y, Meng Y, Xie J, Liu C. Effect of high-energy electron beam irradiation on eating quality of rice. Food Science. 2016; 37(3): 71-4. PubMed PMID: CSCD: 5638309.

4. Zhong F, Li Y, Ibanz AM, Oh MH, McKenzie KS, Shoemaker C. The effect of rice variety and starch isolation method on the pasting and rheological properties of rice starch pastes. Food Hydrocolloids. 2009; 23(2): 406-14. doi: 10.1016/j.foodhyd.2008.02.003. PubMed PMID: WOS: 000260283900019.

5. Du SK, Jiang HX, Ai YF, Jane JL. Physicochemical properties and digestibility of common bean (Phaseolus vulgaris L.) starches. Carbohydrate Polymers. 2014; 108: 200-5. doi: 10.1016/j.carbpol.2014.03.004. PubMed PMID: WOS: 000336193600026.

6. Rani MRS, Bhattacharya KR. Rheology of rice-flour pastes: Relationship of paste breakdown to rice quality, and a simplified Brabender viscograph test. Journal of Texture Studies. 1995; 26(5): 587-98. doi: 10.1111/j.1745-4603.1995.tb00806.x. PubMed PMID: WOS: A1995TR67700008.

7. Vandeputte GE, Vermeylen R, Geeroms J, Delcour JA. Rice starches. III. Structural aspects provide insight in amylopectin retrogradation properties and gel texture. Journal of Cereal Science. 2003; 38(1): 61-8. doi: 10.1016/s0733-5210(02)00142-x. PubMed PMID: WOS: 000183868800006.

8. Bashir K, Swer TL, Prakash KS, Aggarwal M. Physico-chemical and functional properties of gamma irradiated whole wheat flour and starch. Lwt-Food Science and Technology. 2017; 76: 131-9. doi: 10.1016/j.lwt.2016.10.050. PubMed PMID: WOS: 000390726400017.

9. Ai Y, Jane JL. Gelatinization and rheological properties of starch. Starch-Starke. 2015; 67(3-4): 213-24. doi: 10.1002/star.201400201. PubMed PMID: WOS: 000350977700001.

10. Beleia A, Miller RA, Hoseney RC. Starch gelatinization in sugar solutions. Starch-Starke. 1996; 48(7-8): 259-62. doi: 10.1002/star.19960480705. PubMed PMID: WOS: A1996VH67900004.

11. Kohyama K, Nishinari K. Effect of soluble sugars on gelatinization and retrogradation of sweet-patato starch. Journal of Agricultural and Food Chemistry. 1991; 39(8): 1406-10. doi: 10.1021/jf00008a010. PubMed PMID: WOS: A1991GC08700010.

12. Sasaki T, Matsuki J. Effect of wheat starch structure on swelling power. Cereal Chemistry. 1998; 75(4): 525-9. doi: 10.1094/cchem.1998.75.4.525. PubMed PMID: WOS: 000074899600024.

13. Sung WC. Effect of gamma irradiation on rice and its food products. Radiation Physics and Chemistry. 2005; 73(4): 224-8. doi: 10.1016/j.radphyschem.2004.08.008. PubMed PMID: WOS: 000229717200006.

14. Martin M, Fitzgerald MA. Proteins in rice grains influence cooking properties. Journal of Cereal Science. 2002; 36(3): 285-94. doi: 10.1006/jcrs.2001.0465. PubMed PMID: WOS: 000178869300002.

---

## [Decision Letter · Decision Letter 1]

3 Oct 2019

PONE-D-19-15367R1

Characterization of the physical properties of electron-beam-irradiated white rice and starch during short-term storage

PLOS ONE

Dear Dr. Luo,

Thank you for submitting your manuscript to PLOS ONE. After careful consideration, we feel that it has merit but does not fully meet PLOS ONE’s publication criteria as it currently stands. Therefore, we invite you to submit a revised version of the manuscript that addresses the points raised during the review process.

We would appreciate receiving your revised manuscript by Nov 17 2019 11:59PM. To enhance the reproducibility of your results, we recommend that if applicable you deposit your laboratory protocols in protocols.io, where a protocol can be assigned its own identifier (DOI) such that it can be cited independently in the future. For instructions see: http://journals.plos.org/plosone/s/submission-guidelines#loc-laboratory-protocols

We look forward to receiving your revised manuscript.

Kind regards,

Walid Elfalleh, Ph.D

Academic Editor

PLOS ONE

Additional Editor Comments (if provided):

• The abstract should be improved an introductive sentence should be added

• Authors need to add references to the text in introduction section. (ex. Line 35 to 40)

• The whole manuscript should be revised for some syntax and grammar errors before final acceptance

Reviewers' comments:

Reviewer's Responses to Questions

**Comments to the Author**

1. If the authors have adequately addressed your comments raised in a previous round of review and you feel that this manuscript is now acceptable for publication, you may indicate that here to bypass the “Comments to the Author” section, enter your conflict of interest statement in the “Confidential to Editor” section, and submit your "Accept" recommendation.

Reviewer #2: All comments have been addressed

2. Is the manuscript technically sound, and do the data support the conclusions?

Reviewer #2: Yes

3. Has the statistical analysis been performed appropriately and rigorously? 

Reviewer #2: Yes

4. Have the authors made all data underlying the findings in their manuscript fully available?

Reviewer #2: Yes

5. Is the manuscript presented in an intelligible fashion and written in standard English?

Reviewer #2: Yes

6. Review Comments to the Author

Reviewer #2: The authors have addressed all of my questions and concerns, so I have no further questions. Thank you.

7. PLOS authors have the option to publish the peer review history of their article (what does this mean?). If published, this will include your full peer review and any attached files.

Reviewer #2: No

---

## [Author Response · Author response to Decision Letter 1]

14 Oct 2019

We would like to express our sincere thanks to your comments.

Response to Editor Comments

Comments 1: The abstract should be improved an introductive sentence should be added.

Response: An introductive sentence has been added in manuscript. (seen in page 2, line 20-21)

Comments 2: Authors need to add references to the text in introduction section. (ex. Line 35 to 40).

Response: References to the text in introduction section (ex. Line 35 to 40) have been added. (seen in page 3, line 38-41)

Comments 3: The whole manuscript should be revised for some syntax and grammar errors before final acceptance.

Response: We appreciate your kind suggestions. The entire manuscript has been rechecked by a professional editing service. We hope that the revised version would meet the standard of PLOS ONE.

References:

1. FAO. Database: figshare [Internet]. Available from: http://www.fao.org/faostat/en/#data/QC.

2. Zhou Z, Wang X, Si X, Blanchard C, Strappe P. The ageing mechanism of stored rice: A concept model from the past to the present. Journal of Stored Products Research. 2015; 64: 80-7. doi: 10.1016/j.jspr.2015.09.004. PubMed PMID: WOS:000366780900013.

3. Yadav DN, Anand T, Sharma M, Gupta RK. Microwave technology for disinfestation of cereals and pulses: An overview. Journal of Food Science and Technology-Mysore. 2014; 51(12): 3568-76. doi: 10.1007/s13197-012-0912-8. PubMed PMID: WOS:000345916400004.

4. Gregory AG, Toshitaka U, Fumihiko T, Daisuke H. Effect of vapors from fractionated samples of propolis on microbial and oxidation damage of rice during storage. Journal of Food Engineering. 2008; 88(3): 341-52. doi: 10.1016/j.jfoodeng.2008.02.019. PubMed PMID: WOS: 000256815400007

---

## [Editor Report · Decision Letter 2]

21 Oct 2019

PONE-D-19-15367R2

Characterization of the physical properties of electron-beam-irradiated white rice and starch during short-term storage

PLOS ONE

Dear Dr. Luo,

Thank you for submitting your manuscript to PLOS ONE. After careful consideration, we feel that it has merit but does not fully meet PLOS ONE’s publication criteria as it currently stands. Therefore, we invite you to submit a revised version of the manuscript that addresses the points raised during the review process.

We would appreciate receiving your revised manuscript by Dec 05 2019 11:59PM. To enhance the reproducibility of your results, we recommend that if applicable you deposit your laboratory protocols in protocols.io, where a protocol can be assigned its own identifier (DOI) such that it can be cited independently in the future. For instructions see: http://journals.plos.org/plosone/s/submission-guidelines#loc-laboratory-protocols

We look forward to receiving your revised manuscript.

Kind regards,

Walid Elfalleh, Ph.D

Academic Editor

PLOS ONE

Additional Editor Comments (if provided):

The current version of the manuscript seems improved. before finale acceptance, please consider the following comments :

 The authors indicated in the text that there is a significant variation in different parameters however no statistical analysis is done to confirm. Please add ANOVA to figures (figures 1 to figures 4) and specifies if the comparison is made between groups with different irradiation doses or different storage time (days).

---

## [Author Response · Author response to Decision Letter 2]

27 Nov 2019

Response to Editor Comments

Comments 1: The authors indicated in the text that there is a significant variation in different parameters however no statistical analysis is done to confirm. Please add ANOVA to figures (figures 1 to figures 4) and specifies if the comparison is made between groups with different irradiation doses or different storage time (days).

Response: Thank you for your helpful comments. a) We have made some mistakes in using “significant” and “significantly”, and we have revised the content in the manuscript to make the statement more rigorous (seen in page 11, line 169-170 and 180; page 20, line 250-252 and line 257-258; page 21, line 272-273; page 22, line 285-286; page 27, line 338-340; page 35 and 36, line 480-493). b) The ANOVA has been added in Tables S1-4 in supporting files. c) we have revised figure 4 to make it more reasonable and have rechecked the font in other three figures.

---

## [Editor Report · Decision Letter 3]

4 Dec 2019

Characterization of the physical properties of electron-beam-irradiated white rice and starch during short-term storage

PONE-D-19-15367R3

Dear Dr. Luo,

We are pleased to inform you that your manuscript has been judged scientifically suitable for publication and will be formally accepted for publication once it complies with all outstanding technical requirements.

With kind regards,

Walid Elfalleh, Ph.D

Academic Editor

PLOS ONE
---

## [Editor Report · Acceptance letter]

10 Dec 2019

PONE-D-19-15367R3 

Characterization of the physical properties of electron-beam-irradiated white rice and starch during short-term storage 

Dear Dr. Luo:

I am pleased to inform you that your manuscript has been deemed suitable for publication in PLOS ONE. Congratulations! Your manuscript is now with our production department. 

With kind regards,

on behalf of

Professor Walid Elfalleh 

Academic Editor

PLOS ONE